# Functional analysis of Collagen 17a1: A genetic modifier of junctional epidermolysis bullosa in mice

Thomas J. Sproule[1]*, Robert Y. Wilpan[1], Benjamin E. Low[1], Kathleen A. Silva[1], Deepak Reyon[2,3,4,5¤], J. Keith Joung[2,3,4,5], Michael V. Wiles[1], Derry C. Roopenian[1], John P. Sundberg[1,6]

1 The Jackson Laboratory, Bar Harbor, Maine, United States of America, 2 Molecular Pathology Unit, Massachusetts General Hospital, Charlestown, Massachusetts, United States of America, 3 Center for Computational and Integrative Biology, Massachusetts General Hospital, Charlestown, Massachusetts, United States of America, 4 Center for Cancer Research, Massachusetts General Hospital, Charlestown, Massachusetts, United States of America, 5 Department of Pathology, Harvard Medical School, Boston, Massachusetts, United States of America, 6 Department of Dermatology, Vanderbilt University Medical Center, Nashville, Tennessee, United States of America

¤ Current address: Prime Medicine, Cambridge, Massachusetts, United States of America
* tom.sproule@jax.org

**Data Availability Statement:** All relevant data are within the manuscript and its Supporting information files.

**Funding:** This work was supported by grants from DeBRA Austria (DCR) and DeBRA International

## Abstract

Previous work strongly implicated Collagen 17a1 (*Col17a1*) as a potent genetic modifier of junctional epidermolysis bullosa (JEB) caused by a hypomorphic mutation (*Lamc2^jeb*) in mice. The importance of the noncollagenous domain (NC4) of COLXVII was suggested by use of a congenic reduction approach that restricted the modifier effect to 2–3 neighboring amino acid changes in that domain. The current study utilizes TALEN and CRISPR/Cas9 induced amino acid replacements and in-frame indels nested to NC4 to further investigate the role of this and adjoining COLXVII domains both as modifiers and primary risk effectors. We confirm the importance of COLXVI AA 1275 S/G and 1277 N/S substitutions and utilize small nested indels to show that subtle changes in this microdomain attenuate JEB. We further show that large in-frame indels removing up to 1482 bp and 169 AA of NC6 through NC1 domains are surprisingly disease free on their own but can be very potent modifiers of *Lamc2^jeb/jeb* JEB. Together these studies exploiting gene editing to functionally dissect the *Col17a1* modifier demonstrate the importance of epistatic interactions between a primary disease-causing mutation in one gene and innocuous 'healthy' alleles in other genes.

## Introduction

The skin is a tightly interwoven sheath that insulates and protects internal organs from the exterior and is continually renewed to maintain a homeostatic environment [1]. Its overlying epithelium is organized primarily by basal keratinocytes that proliferate and differentiate apically to form the strata spinosum, granulosum and corneum layers. Desmosomes and adherens junctions integrated intracellularly with keratin and actin filaments to confer apical and lateral support. Hemidesmosomes are super-molecular complexes that localize to the base of the basal keratinocyte layer and provide durable connections to the underlying dermis. Plectin

(DCR), National Institutes of Health (NIH) grant number OD011190 (MVW), NIH grant numbers DP1 GM105378 (JKJ) and NIH R01 GM088040 (JKJ), the Jim and Ann Orr MGH Research Scholar Award (JKJ), and by The Jackson Laboratory (DCR). The funders had no role in study design, data collection and analysis, decision to publish, or preparation of the manuscript.

**Competing interests:** TJS, RYW, BEL, KAS, DR, JKJ, MVW, JPS and DCR have no competing interest. JKJ has, or had during the course of this research, financial interests in several companies developing gene editing technology: Beam Therapeutics, Blink Therapeutics, Chroma Medicine, Editas Medicine, EpiLogic Therapeutics, Excelsior Genomics, Hera Biolabs, Horizon Discovery, Monitor Biotechnologies, Nvelop Therapeutics (f/k/a/ ETx, Inc.), Pairwise Plants, Poseida Therapeutics, SeQure Dx, Inc., Transposagen Biopharmaceuticals, and Verve Therapeutics. JKJ's interests were reviewed and are managed by Massachusetts General Hospital and Mass General Brigham in accordance with their conflict of interest policies. JKJ is a co-inventor on various patents and patent applications that describe gene editing and epigenetic editing technologies. This does not alter our adherence to PLOS ONE policies on sharing data and materials.

and the epithelial isoform of dystonin (DST-E/BPAG1/BP230) bind intracellular keratin intermediate filaments and link them to membrane spanning integrin α6ß4 (ITGA6B4) and collagen XVII (COLXVII) whose ectodomains bind extracellular matrix (ECM) proteins. These protein complexes comprise the lamina lucida and extend into the lamina densa forming the basement membrane (BM) and tightly attach to the underlying dermis in ways still only partially understood [2, 3].

Much of the current understanding of this infrastructure has emanated incidentally from the study of rare heritable disorders that weaken and disrupt the skin and other epithelialized surfaces. Epidermolysis bullosa (EB) is the general descriptor for a spectrum of such mechano-blistering disorders in which causality is attributed to defects in any one of at least 20 genes or combinations thereof that compromise the normally durable infrastructure of the cutaneous layers [4–6]. EB can be subtyped based on the plane of the skin affected, with junctional form (JEB) characterized by cleavages at the BM lamina lucida and densa boundaries.

JEB is further distinguished based on severity of phenotype and genetic causality. The most severe forms, which often result in death during infancy or early childhood, are most commonly attributed to loss-of-function mutations in any one of the three genes encoding the laminin α3, ß3 and γ2 subunits of the ECM molecule laminin 332 (L332) [7]. This heterotrimeric complex is secreted by basal keratinocytes and forms a critical bridge between ITGA6B4 heterodimers in the hemidesmosome outer plaque of epidermal basal keratinocytes and collagen VII anchoring fibrils in the underlying papillary dermis. A spectrum of less severe forms, referred to as intermediate JEB, result from less severe mutations (missense; in-frame splicing) of the L332 genes and more commonly from mutations in *COL17A1* [8–13].

*COL17A1* genotype to phenotype analyses of a large cohort diagnosed with JEB-other revealed a pattern in which severe generalized forms concordant with nonsense, insertions and deletions profoundly affect transcriptional stability and/or translation, resulting in a virtual absence of COLXVII and gross hemidesmosomal disruptions [9, 14]. Similarly, mice genetically lacking *Col17a1* survive to birth but succumb within weeks thereafter from hemidesmosomal disruptions and severe blistering syndrome [15]. Less crippling *COL17A1* mutations are causally associated with more attenuated and highly variable phenotypes, often referred to as localized and/or late onset JEB-other [16]. This includes splice site mutations that may diminish but do not negate COLXVII protein and a missense change (R1303Q) [17]. A major product of basal keratinocytes, COLXVII (also known as BPAG2 and BP180) is a conserved 180 kDa type II transmembrane protein. It forms a homotrimer whose 60 kDa globular intracellular domain interacts with the epithelial isoform of dystonin (DST-E), plectin and integrin β4 to link the keratin cytoskeleton to hemidesmosomes [2]. Its 120 kDa extracellular domain (1008 AA in humans, 972 AA in mice) is organized into alternating collagenous (Col) and non-collagenous (NC) subdomains (15 Col and 16 NC in humans, 14 Col and 15 NC in mice), whose functions have been challenging to define in a physiological context [18, 19].

Solid-phase biochemistry, cell binding assays and mutational analyses are consistent with a supportive role for the ectodomain of COLXVII (ectoCOLXVII) in epidermal homeostasis at the macro level. Specifically, ectoCOLXVII is proposed to strengthen basal keratinocyte adhesions to the underlying BM through its binding to L332 [20–22]. Findings that many *COL17A1* mutations that are causally associated with nH-JEB localize to exons 50 and 51 (encoding the distal NC5-Col4-NC4-Col3-proximal NC3 subdomains) are thus consistent with a functional interaction of these subdomains in binding L332. Moreover, there is evidence that ectoCOLXVII can be further enzymatically cleaved into a shorter 97Kd N-terminal fragment. This shed product has been suggested to promote keratinocyte mobility that is necessary for correct BM formation [23].

Mouse models provide a useful surrogate to decipher the genetic and functional architecture of JEB as well as other forms of EB. A neonatal lethal null mutation of the laminin gamma 2 gene (*Lamc2^tm1Uit*) is too severe to investigate modifier genes but a spontaneously-arising, hypomorphic mutation (*Lamc2^jeb*) is quite effective for this purpose [14]. *Lamc2^jeb* homozygous mutants survive into adulthood but demonstrate weakened dermal–epidermal adhesion, substantially reduced levels of L332 in the BM and progressive clinical symptoms that are a remarkable phenocopy of human intermediate JEB. In this model, genetics distinct from *Lamc2* play a large role in determining the severity of JEB [14], illustrating the potential of genetic modifiers that may help account for the considerable phenotypic variation within subtypes of EB and within affected families [24]. One such modifier is *Col17a1*. The effect was reduced to an interval carrying two candidate B6/PWD missense single nucleotide polymorphisms (SNPs) in exon 50 that encode p1275 S/G and p1277 N/S in the NC4 domain of COLXVII [25]. These changes neighbor a rare missense human mutation R1303Q (mouse p1282) that is causally associated with nH-JEB/JEB-other in humans. This underscores the potential importance of subtle alterations in NC4 and adjoining subdomains of COLXVII acting through L332 in supporting junctional dermal–epidermal adhesion, BM maintenance and causing JEB.

Here, we utilized gene editing methods in mice to further investigate the role of COLXVII in epidermal homeostasis and the pathogenesis of JEB. The phenotypic consequences of a series of nested and tiled in-frame indels and missense substitutions that map to mouse C-terminal NC and Col subdomains of COLXVII were analyzed in the context of wild-type (WT, *Lamc2^wt/wt*) and *Lamc2^jeb/jeb* mutant mice. Homozygous *Col17a1* mutations on an inbred wild-type B6 background (*Lamc2^wt/wt*) mice proved surprisingly free of disease phenotypes. However, several of these mutations acted to potently impact dermal-epidermal integrity and JEB in mice whose epidermis is compromised by homozygosity for the *Lamc2^jeb* mutation. Together these studies demonstrate the utility of gene editing toward the functional dissection of complex proteins *in vivo* and emphasize the importance of genetic factors distinct from the primary risk mutation in mechanobullous disorders.

## Results

### Ultrafine recombinant mapping and gene editing identify allelic changes limited to two neighboring AA in the NC4 domain of COLXVII that explain the *Col17a1* B6/PWD modifier of *Lamc2^jeb*-induced JEB

Previous studies used recombination mapping of B6-*Lamc2^jeb/jeb* Chr19^PWD/PWD congenic mice to limit a potent PWD modifier of *Lamc2^jeb*-induced JEB to a short (1085 bp) interval including exon 50 of *Col17a1* [25]. (The unusually high level of genetic resolution was likely facilitated by a prototypic *Pdrm9* recombination motif hotspot centered in this fragment [26]). This interval included 3 missense SNPs encoding B6/PWD amino acid polymorphisms of p1275 S/G, p1277 N/S and p1292 T/I within the NC4 and Col3 domains of COLXVII (Fig 1A and 1B). Here reanalysis and correction of R03L congenic limits further reduction of the candidate interval to 723 bp. This interval includes the missense SNPs encoding p1275 and p1277 PWD alleles in addition to a number of synonymous and intronic SNPs but excludes the p1292 encoding SNP (Fig 1A).

To more directly address the candidacy of the p1275 and p1277 PWD SNPs, Transcription Activator-Like Effectors (TALENs) were used to introduce germ line edits in the *Col17a1* locus. TALENs designed to cause double-stranded breaks at p1275 coupled with an ssDNA template designed to introduce the p1275 S→G and p1277 N→S SNP changes were injected into B6 zygotes (Figs 1B and 3A). Among the mutations identified and propagated, one

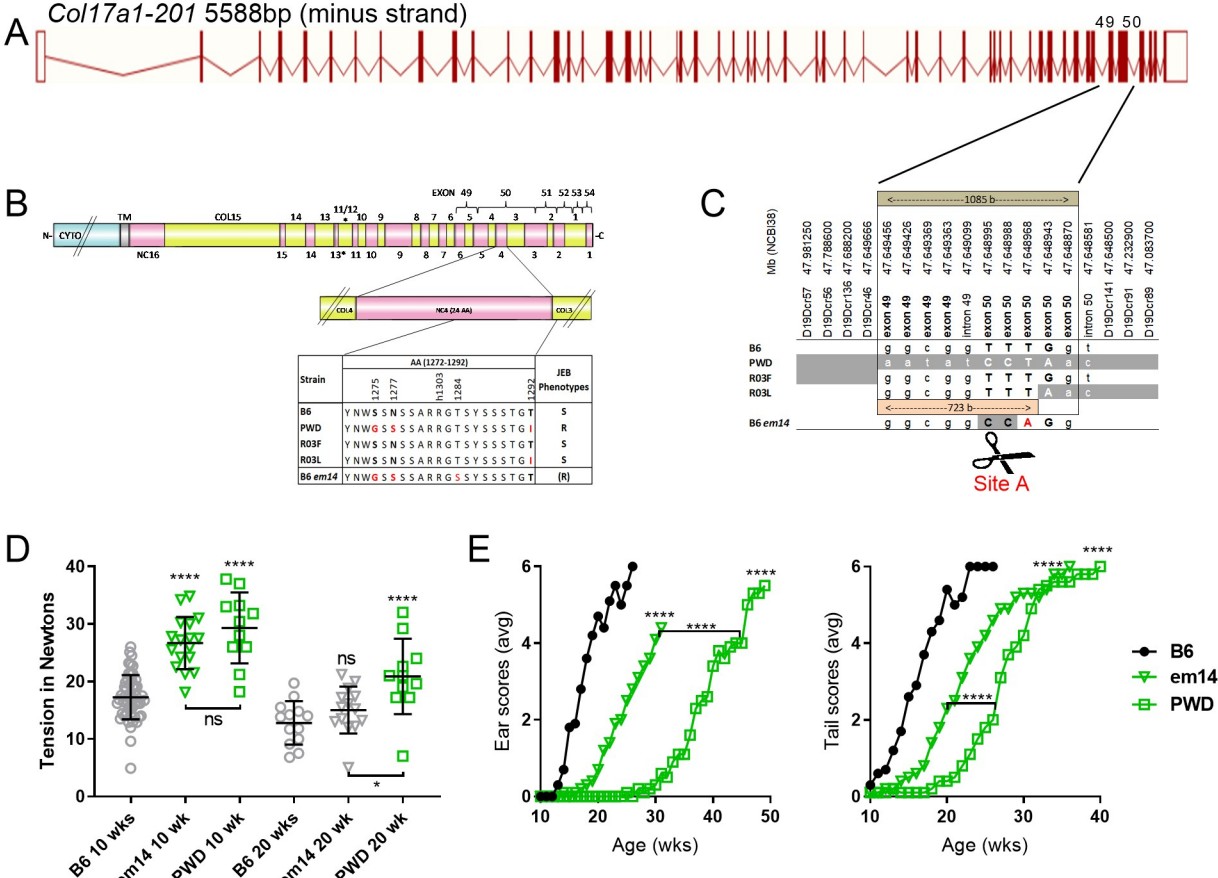

**Fig 1. Congenic recombinants and TALEN amino acid replacement.** (A-C) B6.PWD updated DNA and protein maps including *em14* TALEN induced replacement. (A) Image of the longest and most often discussed mouse isoform of *Col17a1*, Col17a1-201, from Ensembl (ensembl.org) showing position of relevant polymorphisms (C). B6.PWD and R03F are from previous work, R03L limits corrected from previous work [25]. B6 allele white, PWD allele gray. Previous candidate interval 1085 bp. Revised candidate interval 723 bp based on corrected R03L limits. Polymorphic exonic SNPs are bold if they result in a missense change. In (B), * indicates Col and NC region names skip NC12 and use hybrid name Col11/12 to maintain mouse region numbering aligned to comparable human regions (Fig 2). Phenotypes S = susceptible, R = resistant. (D) 10 and 20 week male *em14* tension compared to controls. (E) *Em14* ear and tail scores compared to controls. All data shown (D-E) is for males homozygous *Lamc2^jeb/jeb* and *Col17a1^em14/em14* or indicated congenic segments. All lines are B6-*Lamc2^jeb/jeb* 'long congenic' as used previously [25]. Statistics are compared to B6 unless otherwise indicated (tension 10 wk vs 10 wk, 20 wk vs 20 wk), ns if not significant. Tension based on 1-way ANOVA of every set vs every other set. Ear and tail score p values are pairwise survival comparisons of each strain to B6 or as indicated based on age in weeks scores first '4' (less significance of Log-rank (Mantel-Cox) or Gehan-Breslow-Wilcoxon test results indicated).

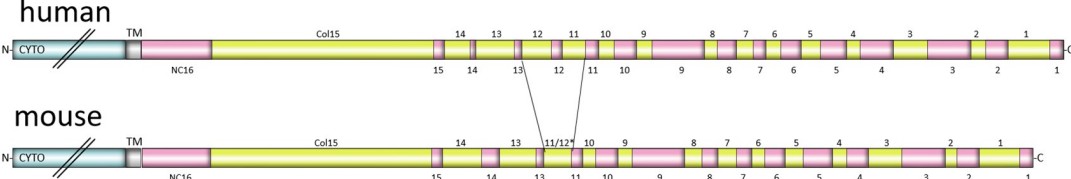

**Fig 2. Alignment of human and mouse extracellular collagen XVII.** Mouse full protein is 27AA shorter than human. Mouse extracellular is 36AA shorter than human and has one less NC and one less Col domain than human. Mouse NC and Col segments are numbered here to best align with their human counterparts. Mouse Col "11/12*" includes a proximal amino acid region similar to human Col12 and distal region similar to human Col11.

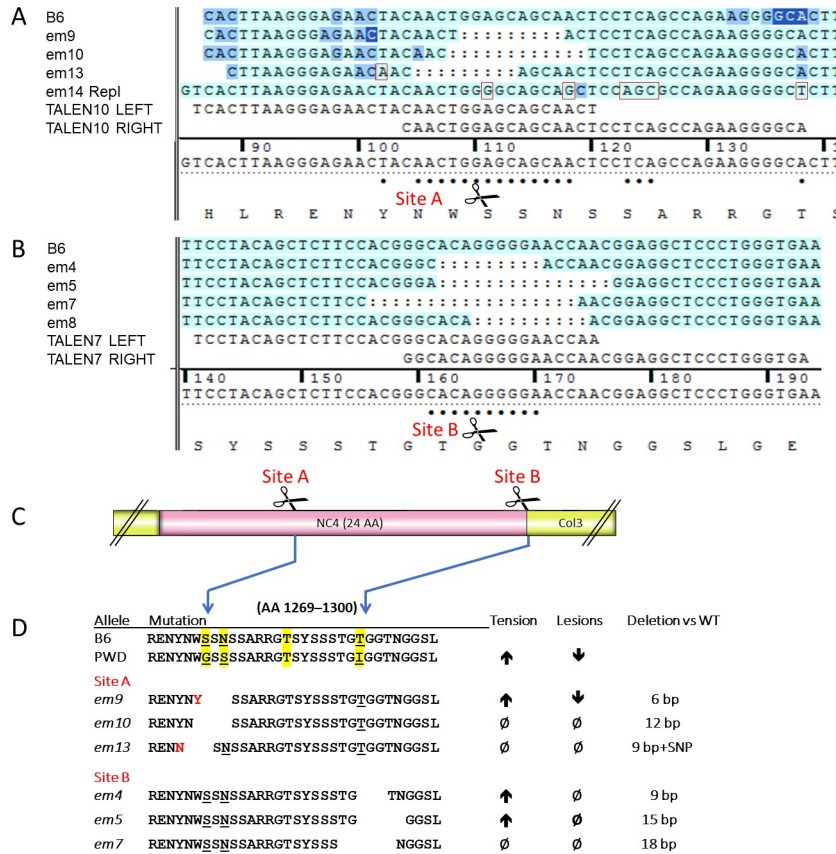

**Fig 3. Small in-frame deletions sequence.** (A) DNA sequence showing deletions and changes at cut site A including for *em14* replacement from Fig 1. (B) DNA sequence deletions and alterations at cut site B including for em8 frame shift deletion with results in Figs 5–7. (C) Cut sites in relation to protein regions NC4 and Col3. (D) Changes in amino acid sequence resulting from DNA changes.

founder line (*Col17a1*^em14Dcr^, henceforth referred to as *em14*) was predicted by DNA sequencing and confirmed by cDNA sequencing to create the desired p1275 S→G and p1277 N→S transpositions. However, the targeting event also created an inadvertent missense SNP at the 3' terminus of the guide site that was predicted to introduce a novel p1284 T→S change (Figs 1A, 1B and 3A).

*Em14* homozygotes on a B6 background (B6-*Lamc2*^wt/wt^ *Col17a1*^em14/em14^) did not exhibit any abnormalities up to 37 weeks of age (Fig 4B). To determine if the *em14* substitution could modify JEB, the *em14* allele was introduced into B6-*Lamc2*^jeb/jeb^ mice (henceforth called *em14 +jeb*), to create compound homozygous mice. These were monitored for dermal-epidermal adhesion and ear and tail lesion development. Tail tension test values for *em14+jeb* were similarly high and equivalent to the fully protective *Col17a1*^PWD^ haplotype compared with susceptible B6-*Lamc2*^jeb/jeb^ mice at 10 wks of age (p < .0001), an age before ear and tail lesions were apparent, but did not achieve levels found with the PWD allele at 20 wks (Fig 1C). Similarly, longitudinal ear and tail lesion scoring indicated that the *em14* was intermediate compared with the B6 and PWD alleles (Fig 1D). The results support the key involvement of the p1275 S→G and p1277 N→S changes in the NC4 COLXVII subdomain but also suggested that penetrance of this modifier can be partially attenuated by the inadvertent downstream p1284 T→S substitution.

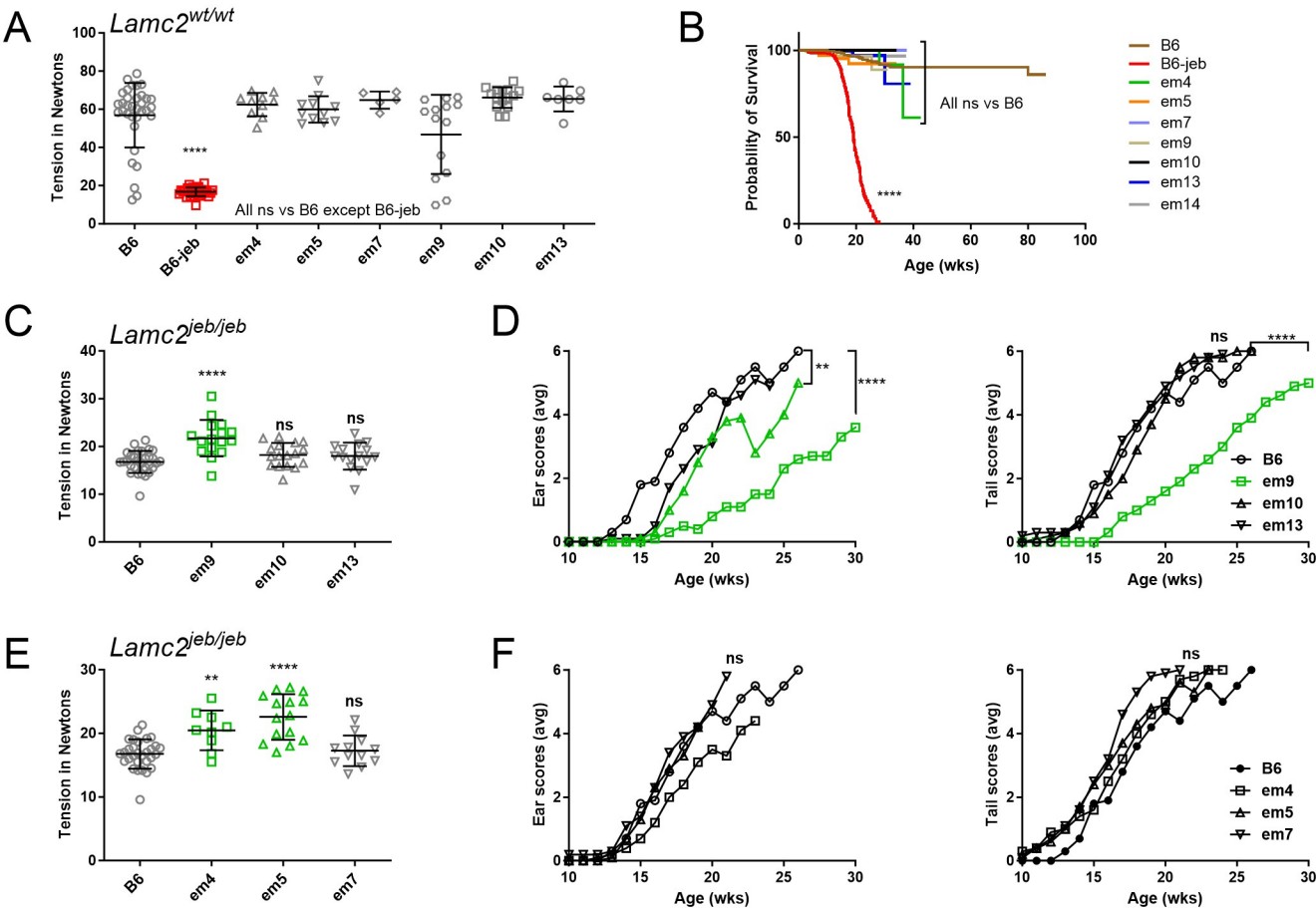

**Fig 4. Small in-frame deletion tension and survival results.** (A-B) Tension (A) and censored survival cause of death (B) of all IFD on a *Lamc2^{wt/wt}* background (*e.g.* B6-*Lamc2^{wt/wt} Col17a1^{em4/em4}*). (A) is all males. Survival is of all mice (both sexes) from colony during period of IFD breeding, showing percent survival decrease for suspect 'causes of death' such as 'sore', 'dermatitis', 'dead' and 'missing' and censored values for 'ID' (inventory discard with no notable conditions). Mice deliberately removed for experiments or other reasons are removed from consideration here. Survival graph includes replacement *em14* (B6-*Lamc2^{wt/wt} Col17a1^{em14/em14}*). (B) all *Lamc2^{wt/wt}* except 'B6-jeb' is B6- *Lamc2^{jeb/jeb}* included as a control. (C-D) Tension (C) and ear and tail scores (D) for three in-frame deletions at cut site A. (E-F) Tension (E) and ear and tail scores (F) for three in-frame deletions at cut site B. All data shown in (C-F) is for males homozygous *Lamc2^{jeb/jeb}* and for indicated *Col17a1^{em/em}* (*e.g.* B6-*Lamc2^{jeb/jeb} Col17a1^{em4/em4}*). Tension statistics are 1-way ANOVA and ear and tail score statistics survival based on age to reach score '4' ('moderately affected'), as in Fig 1.

## Some indels nested to the *Col17a1* modifier microdomain attenuate *Lamc2^{jeb}*-induced JEB

To further inquire into the functional involvement of the p1275/1277 and p1292 sites, the indels that often arise collaterally as a byproduct of targeted gene editing were utilized. Six founder lines, 3 each from cut sites A (p1275, lines *em9*, *em10* and *em13*) and B (p1294, lines *em4*, *em5* and *em7*) predicted by DNA sequencing to carry small (9–18 bp) in-frame indel nesting the sites of modifier variation were propagated (Fig 3A and 3B). Each was made homozygous on a B6 (*Lamc2^{wt/wt}*, named *em9*, etc.) and B6-*Lamc2^{jeb/jeb}* (named *em9+jeb*, etc.) background and evaluated for standalone and modifier effects. Mice homozygous for the *Col17a1* mutations without *Lamc2^{jeb/jeb}* were born healthy, bred normally and did not exhibit dermal-epidermal weakness as measured by tail tension test at 10 weeks of age or epidermal lesions or other observable deleterious manifestations to at least to 30 wks of age (Fig 4A and 4B). In

modifier testing on a B6-*Lamc2*$^{jeb/jeb}$ background, *em4+jeb*, *em5+jeb* and *em9+jeb* all statistically improved skin adhesion strength as measured by tail tension test of 10 wk old males, though only *em9+jeb* coupled that with delay of lesion onset as measured by ear and tail scores. *Em10+jeb* showed improvement by subjective ear scores but not tail scores or tension. *Em7 +jeb* and *em13+jeb* did not alter standard *Lamc2*$^{jeb/jeb}$ phenotypes in any way (Fig 4C–4F). No novel phenotypes which could be attributed to digenic interaction of the two mutations were detected.

To confirm that the effects of the *em9* mutation on COLXVII were discrete, capillary immunoelectrophoresis was performed using neonatal skin from an *em9* homozygote. Significant changes in size or abundance of immunoreactive COLXVII products compared to wild-type B6 skin COLXVII were not apparent when detected with a rabbit antisera (Atlas HPA043673) reactive to epitope(s) in the intracellular domain or by a rabbit monoclonal antibody (Abcam ab184996) reactive to an unspecified epitope within the NC4 to Col2 fragment of COLXVII (Fig 5D and 5E).

In addition to the in-frame indel, we also recovered a single small (10 bp) deletion founder stock (*em8*) predicted by genomic and cDNA analyses to cause a frame shift and translational stop (Fig 3A). Unlike other mutations acquired for this study, *em8* proved to be homozygous lethal. *Em8/em8* pups were similar in size and appearance to littermates at birth but were notably smaller and less healthy beginning at ~5 days old and typically died at 7–17 days of age (Figs 6D and 7B). Histological examination of *em8* homozygotes at 6, 10 and 27 days of age compared to littermate controls revealed extensive dermal-epidermal separation (Fig 7A) comparable to human collagen XVII deletion induced JEB and a previously published mouse *Col17a1* knockout model [15]. Twelve litters born to B6-*Col17a1*$^{em8/wt}$ het x het intercross matings were tracked from 1 to 21 days old and genotyped at 1–4 days old and again at >21 days old. They included 20 *wt/wt* (29.0%), 35 *em8/wt* (50.7%) and 14 *em8/em8* (20.3%). All *em8/em8* hom from these litters died by 17 days of age. Nearly normal Mendelian ratios indicate most *em8/em8* induced lethality is post-birth.

## Identification and characterization of mice carrying large *Col17a1* indels

Phenotypic consequences of deletions that eliminate larger fragments of NC4 and adjoining COLXVII subdomains were investigated. Founders from projects with four different TALEN or CRISPR/Cas9 cut sites within *Col17a1* exon 50 or adjacent intron 50–51 were PCR tested with appropriate primer combinations to detect large deletions at each site. Six large indels were identified and bred, all of which proved to be viable in the homozygous state. *Em1* excised a 567 bp sequence including all of exon 50 and portions of its flanking introns. *Em2* excised a 650 bp fragment extending from intron 48–49 to mid-exon 50, overlapping the proximal end of the *em1* deletion. The remaining four indels, all derived from the CRISPR/Cas9 cut site C (Fig 5A), all overlap the distal portion of the *em1* deletion. *Em15* deleted 2 fragments totaling 1482 bp–distal exon 50 through mid-exon 53 and intron 53–54 into the 3'UTR. *Em16* deleted 1449 bp from distal exon 50 through mid-intron 53–54, replacing it with a 6 bp insertion. *Em17* deleted 140 bp from distal exon 50 to mid-intron 50–51. *Em18* deleted 2 bp in distal exon 50 but inserted 168 bp in its place, composed of two stretches of novel DNA (30 and 11 bp respectively) flanking a 127 bp segment matching *Col17a1* exon 53 and its flanking intronic sequence (Figs 5A, 8 and 9). Sequencing of cDNA PCR products revealed *em1* deleted the entire 120AA region encoded by exon 50 (mid-NC5 through mid-NC3); *em2* deleted the entire 169AA encoded by exons 49 and 50 (mid-NC6 through mid-NC3); *em15* disrupts mid NC3, inserting 13 novel AA followed by a stop, and is thus missing the entire distal end of the protein (114AA, mid-NC3 through NC1); *em16* produces two cDNA products, the first of which

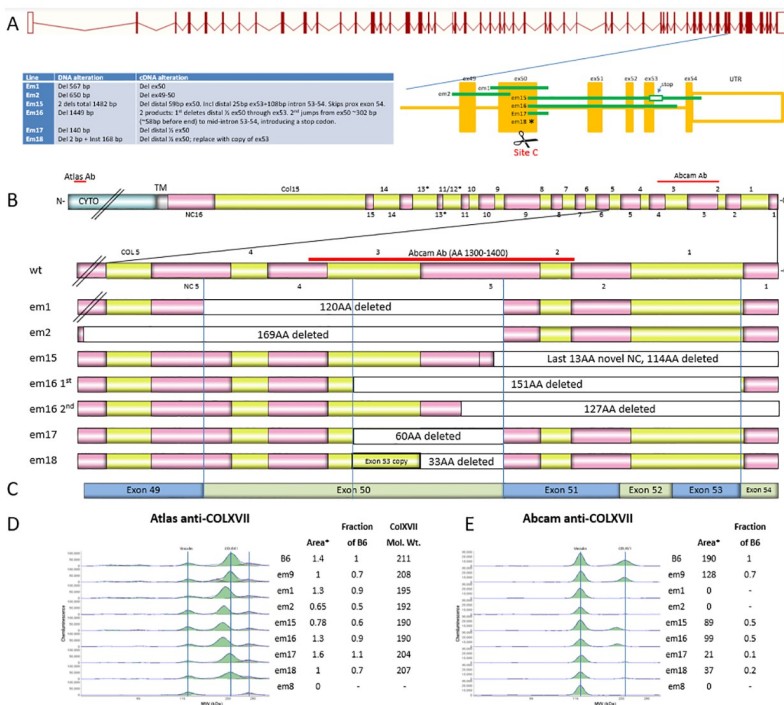

**Fig 5. Large indel DNA and protein effects.** (A) Col17a1-201 genetic locations and sizes of *em1*, *em2* and *em15-18* large indels. Indicated CRISPR cut site C resulted in *em15-18*. (B) Mouse collagen XVII protein sequence, extracellular full-length top expanded beneath to show protein regions deleted by each mutation. Vertical lines are to clarify alignment. (C) Distal *Col17a1* cDNA structure aligned to protein structure in (B). (D) Neonatal (3–4 day old) trunk skin protein detection by capillary immunoelectrophoresis using two anti-COLXVII antibodies with map positions shown in (B). Area of COLXVII peak and fraction of B6 area normalized to vinculin loading control indicated. Area values given are divided by 1e3 for Abcam and 1e6 for Atlas. COLXVII peak molecular weight included for Atlas antibody reactive against the cytoplasmic domain. N = 1 mouse tested per line.

excises 151AA from mid-NC3 through most of NC1 and the second introduces a stop midway through exon 50 NC3, deleting the last 127AA of the protein; *em17* deletes 60AA encoded by the latter half of exon 50 (mid-Col3-mid-NC3) and *em18* deletes a segment identical to *em17* but inserts a Col segment duplicating the exon 53 portion of Col1 in its place, resulting in loss of proximal NC3 but retention of a Col hybrid segment of proximal Col3 and copy of exon53 Col1 which is almost the same size as, and may serve functionally like, native Col3 (Figs 5B and 9).

To directly assess the consequences of these *Col17a1* deletions and indels at the protein level, capillary immunoelectrophoresis on trunk skin was done using extracts from neonatal mice homozygous for each mutation (Fig 5D and dE). Atlas antisera specific for the intracellular domain revealed single products of reduced sizes consistent with predictions from the DNA and cDNA analyses was detected for all mutations tested, with the exception of the frame-shift *em8* mutant that lacked immunoreactive product (Fig 5D). An Abcam monoclonal antibody specific for an unspecified epitope within the NC4 to NC2 interval failed to detect products from *em1* and *em2* and minimally detected products from *em17* and *em18* (Fig 5E) which, compared to the Atlas results, confirms production of a protein but deletion of an interval in each. Weak peaks for COLXVII for *em17* and *em18* suggest the Abcam antibody binds partly to the region deleted in both and partly to regions retained in both (Fig 5B). Lack of *em8*

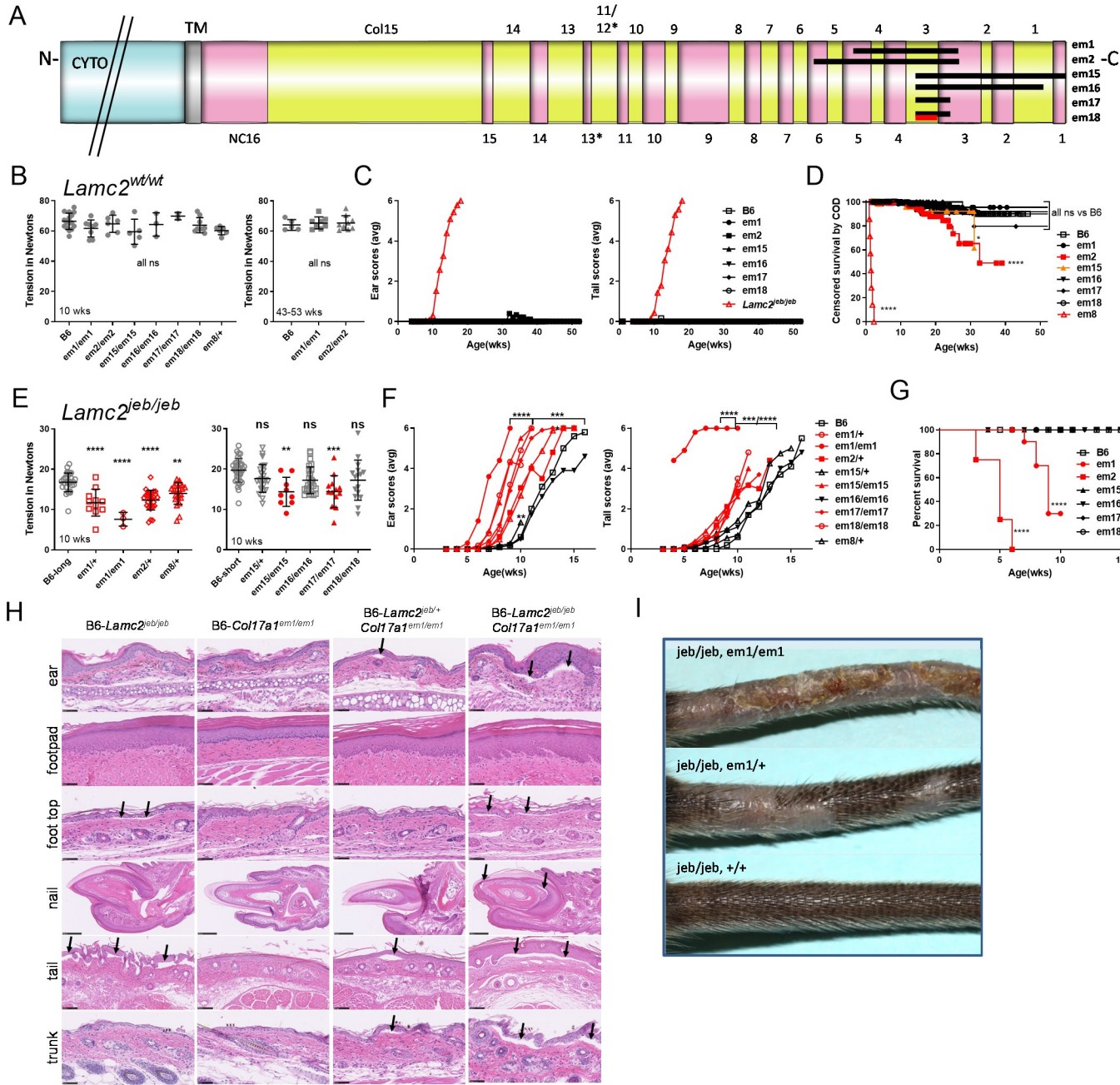

**Fig 6. Large indels and *em8* frame shift.** (A) Map showing alignment of large deletions and *em18* insertion to COLXVII protein (simplified from Fig 5B). (B-D) Phenotypes when indel on B6-*Lamc2^wt/wt^* background, all homozygous *Col17a1^em/em^* except indicated *Col17a1^em8/+^* heterozygote in (B): (B) 10 and 43–53 week old tension, (C) ear and tail lesion/condition scores with B6-*Lamc2^jeb/jeb^* included as a control, (D) censored survival based on 'cause of death' to 1 year of age as in Fig 4B. (B-C) all males. (D) mixed sexes. (E-G) Phenotypes when indel on B6-*Lamc2^jeb/jeb^* background (zygosity of indels indicated in (E-F), all homozygous *Col17a1^em/em^* in (G)): (E) 10 week old tension, (F) ear and tail scores, (G) survival to 15 wks of age. (E) is split to compare each *em* to correct controls (B6-*Lamc2^jeb/jeb^* long or short congenic) that they were crossed to and tension tested during the same period. (E-G) All males. (H) Hematoxylin and eosin evaluation of ~29 day old males comparing B6-*Lamc2^jeb/jeb^* and B6-*Col17a1^em1/em1^* parental strains to B6-*Lamc2^jeb/+^ Col17a1^em1/em1^* and B6-*Lamc2^jeb/jeb^ Col17a1^em1/em1^*. Arrows point to regions of dermal-epidermal separation. (I) Photo of phenotype differences for *em1/em1* vs *em1/+* vs +/+ at 10 weeks of age from (G).

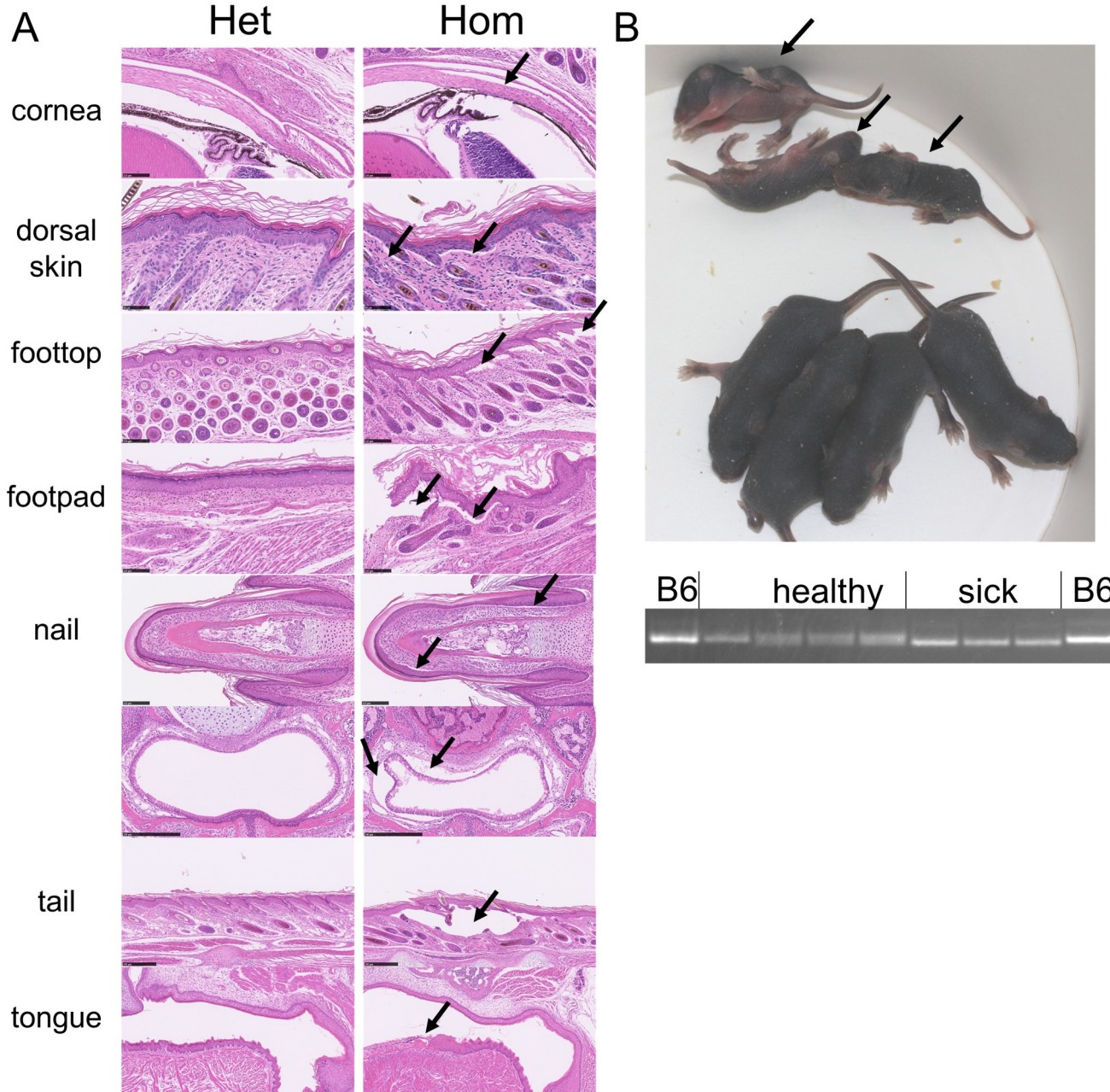

**Fig 7. *Em8* het vs hom 6 day old comparisons.** (A) Hematoxylin and eosin staining of multiple tissues showing dermal-epidermal splits (arrows) in homozygote but not het control. Bars are 50 μm for dorsal skin, 100 μm for cornea, dorsal foot skin (foot top) and foot pad and 250 μm for all others. (B) Photo and PCR results for 6 day old litter from *em8* het x het mating showing 3 sick pups (arrows) are homozygous for deletion and 4 healthy pups are het or homozygous wild-type. All *Lamc2*$^{wt/wt}$.

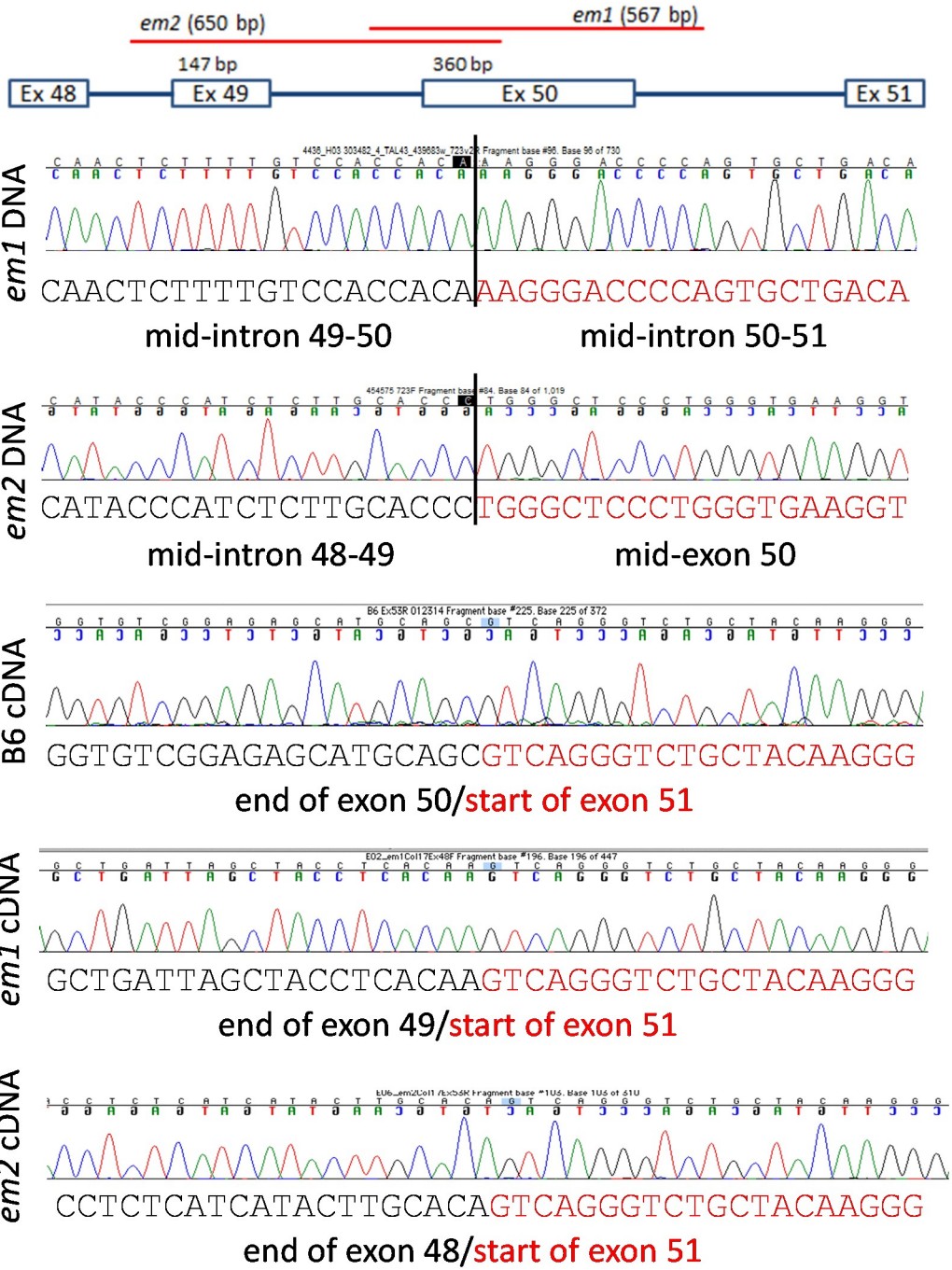

**Fig 8. DNA and cDNA sequence of *em1* and *em2*.**

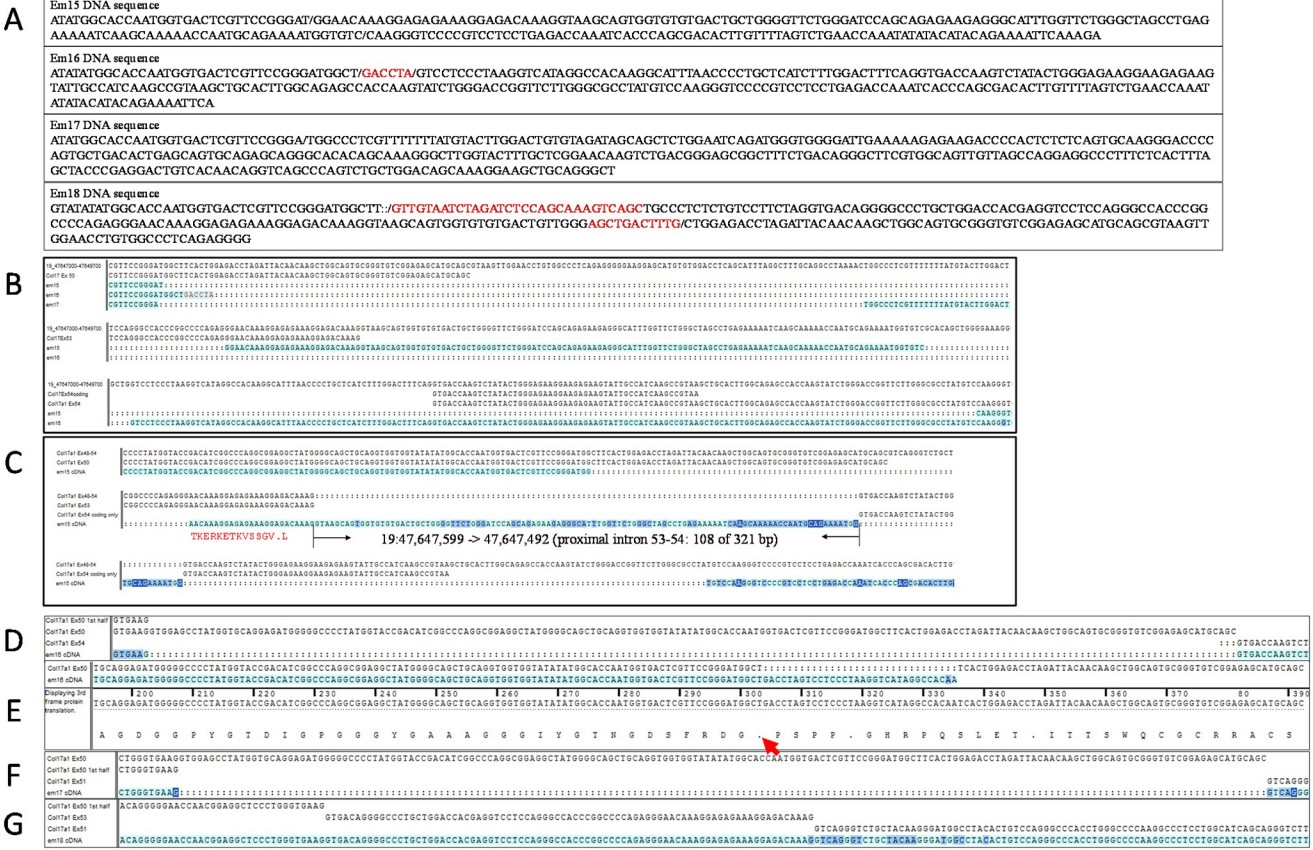

**Fig 9. *Em15-18* DNA and cDNA sequence.** (A) Sequence from *em15*, *em16*, *em17* and *em18* homozygous DNA samples. / indicates breaks in alignment to genomic sequence. Red is novel insertions. Insertions in *em18* bracket a copy of region containing exon 53, 19:47647574–47647700 in Ensembl build 38. (B) *Em15*, *em16* and *em17* DNA aligned to mouse *Col17a1* genomic sequence. (C) Alignment of *em15* homozygous cDNA sequence with *Col17a1* exons. A portion of *em15 cDNA* sequence immediately distal to exon 53 maps to proximal intron 53–54. cDNA distal to exiting exon 50 codes as TKERKETKVSSGV.L (13 novel AA then stop). (D) *em16* first cDNA product skips from mid-exon 50 to the start of exon 54. (E) *em16* 2nd cDNA product introduces an immediate stop (arrow) upon deviating from exon 50. Deviation sequence maps to distal intron 53–54. (F) *em17* cDNA skips distal half of exon 50, starts coding again at beginning of exon 51. (G) *em18* cDNA aligned with proximal half of exon 50 (skips distal half), then copy of exon 53, then exon 51.

product detectable by the Abcam antibody is likely confirmation of the Atlas antibody results rather than due to disruption of the Abcam monoclonal antibody binding site, though that cannot be completely ruled out.

## Mice are tolerant to mutations that substantially excise or alter C-terminal COLXVII subdomains

All six large indels (*em1*, *em2*, *em15-18*) resulted in mice that were fertile, bred normally and developed healthy progeny in numbers that were not distinguishable from B6 controls when maintained homozygous on a B6 background (S1 Table). Tension tests of homozygotes at 10 and 43–53 weeks of age failed to remove skin and gave values similar to controls, indicating no skin fragility (Fig 6B). Longitudinal ear and tail lesion scoring of homozygotes were normal to at least 44 weeks of age, revealing no blisters or erosions (Fig 6C). Censored survival analysis based on cause of death (COD) notes revealed a significant percentage of affected *em2* compared to B6 controls, mostly noted as 'sore' or 'dermatitis'. By this analysis method *em15* also

reached statistical significance, but review of the primary data suggests that to be an artifact (see supporting information). *Em1*, *em16*, *em17* and *em18* do not statistically differ from B6 (Fig 6D). Histologically, about half of nine *em2/em2* evaluated at 42–44 weeks of age by comprehensive necropsy and skin samples from one of two *em15/em15* at 53 weeks of age exhibited occasional mild ulcers and lesions unlike epidermolysis bullosa but similar to reported B6 strain specific alopecia and dermatitis [27]. None of four *em1/em1* or thirteen B6 controls examined by comprehensive necropsy at 42–44 weeks of age or skin samples from four *em16/em16* at 52 weeks of age included the same defects (Figs 6H and 10), suggesting that *em2* and *em15* deletions may mildly compromise dermal-epidermal adhesion, not enough to cause epidermolysis bullosa, but enough to increase susceptibility to alopecia and dermatitis to which B6 are prone [27]. Taken together, the results indicated that the interval spanning the NC6 to NC1 subdomains of COLXVII are not required for normal cutaneous homeostasis or other life supporting functions (summarized in Table 1).

B6 substrains are prone (as a strain specific lesion) to develop what is known as B6 alopecia and dermatitis or B6 ulcerative dermatitis due to hypomorphic mutations in alcohol dehydrogenase 4 (*Adh4*). Follicular dystrophy is a critical but not always seen part of the lesion. It can result in dystrophy and hair shafts burrowing through the walls of hair follicles inducing an

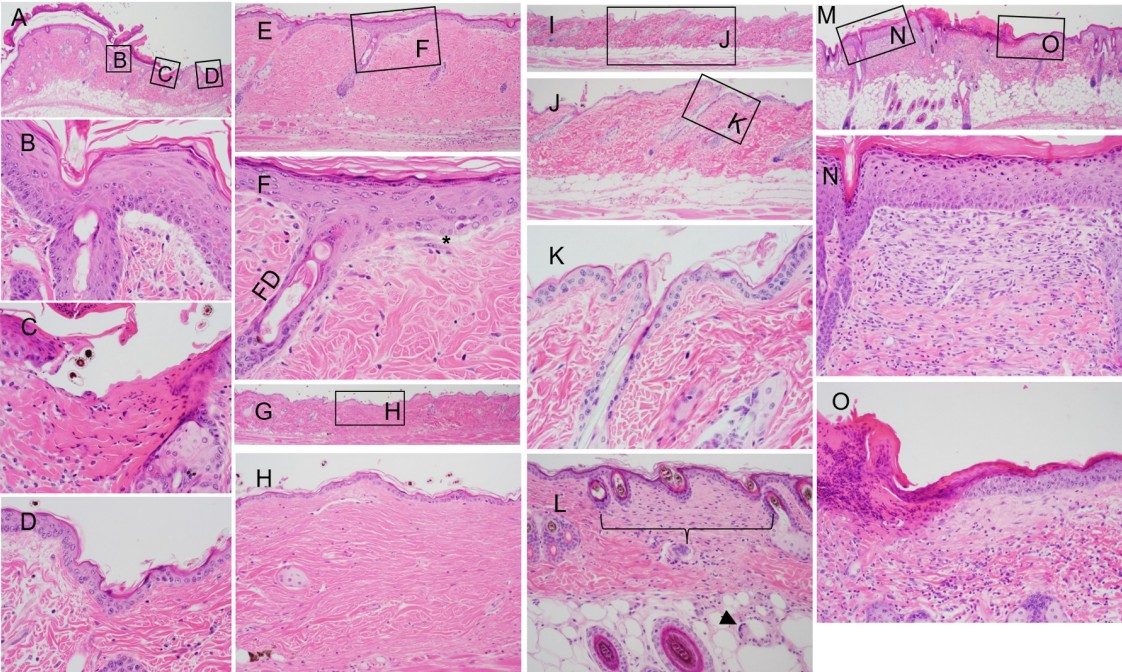

**Fig 10. Large deletion histology (all *Lamc2*<sup>wt/wt</sup>).** (A-H) *Em2* (B6-*Col17a1*<sup>em2/em2</sup>) male 43 weeks of age. (A) Locally extensive area of epidermal hyperplasia covered by degenerating inflammatory cells (mostly neutrophils) and desiccated proteinaceous material (scab). This covers an area of very mild dermal-epidermal separation (B) adjacent to a small ulcer (C) with transition to normal epidermis (D). (E, magnified in F) Focal area of epidermal hyperplasia overlying follicular dystrophy (FD) adjacent to a small area of dermal-epidermal separation (*). (G, magnified in H) A small area of dermis with fine dense regular collagenous connective tissue devoid of hair follicles (old mature scar) adjacent to more normal dense irregular collagenous connective tissue. Note the pigmentary incontinence (arrow, remnant of a dystrophic hair follicle). (I magnified in J and K) Normal skin from B6 wild type male mouse 43 weeks of age. Not the thin epidermis, tight dermal-epidermal junction, normal hair follicle, and uniform dense irregular collagenous connective tissue making up the dermis. (L) *Em16* 52-week-old male. *Em16* rarely had foci of fibrosis (brackets) with underlying granulomatous inflammation. Note the multinucleated giant cell (arrow). (M magnified in N and O) *Em15* male 52 weeks of age. Cutaneous ulcer and scar. (N) Epidermal hyperplasia overlying an area of fibrosis (scar formation). (O) Ulcer (left) with overlying scab formation. Mild fibrosis under mild epidermal hyperplasia with a mixed dermal inflammatory cell infiltrate.

**Table 1. Monogenic and epistatic effects of *Col17a1* mutations.**

| Mutation | Effect on COLXVII protein | Monogenic (vs *Col17a1* WT) | | | Digenic (vs *Col17a1* WT / *Lamc2*$^{jeb}$) | | |
|---|---|---|---|---|---|---|---|
| | | Tension @ 10 wks | Lesions | Survival to 53 wks | Tension @10 wks | Lesions | Survival to XXX wk |
| *em1/+* | DelNC4-XX | ND | ø | Ø | ↓ | ↑ | ND |
| *em1/em1* | | ø | ø | Ø | ↓↓ | ↑↑ | ↓↓ |
| *em2/+* | DelNC4-XX | ND | ø | Ø | ↓ | ND | ND |
| *em2/em2* | | ø | ND | Ø | ND | ND | ↓↓↓ |
| *em15/+* | DelNC4-XX | ND | ø | Ø | ø | ø/↓ | ND |
| *em15/em15* | | ø | ø | Ø | ↓ | ↓ | ø |
| *em16/em16* | Etc | ø | ø | Ø | ø | ø | ø |
| *em17/em17* | | ø | ø | Ø | ↓ | ↓ | ø |
| *em18/em18* | | ø | ø | Ø | ø | ø | ø |
| *em8/+* | Absent | ø | ND | Ø | ø | ø | ø |
| *em8/em8* | | ND | ND | ↓↓↓↓ | ND | ND | ND |

acute and then granulomatous reaction that can lead to trauma (scratching) and ulcerations which can become severe and extensive. The lesions can be difficult to distinguish from ulcers that result from rupture of vesicle and blisters as arise in the various forms of EB. Very minor loss of dermal epidermal adhesion may predispose older mice to develop these lesions associated with B6 dermatitis [27].

## Large deletions nested to the NC4 subdomain most potently exacerbate JEB in *Lamc2*$^{jeb}$ mice

Similar to *em14* replacement and small in-frame deletions reported above, mice with all six large in-frame indels were crossed to B6-*Lamc2*$^{jeb/jeb}$ mice to produce double homozygotes (*e. g.* B6-*Lamc2*$^{jeb/jeb}$ *Col17a1*$^{em1/em1}$, henceforth named *em1+jeb*, etc.) to test for modification of *Lamc2*$^{jeb/jeb}$ induced JEB disease or unique digenic effects by gross observation and standard tail tension, ear score and tail score measurements (Fig 6E–6G). Combined phenotypes, including survival in the case of *em1+jeb* and *em2+jeb* double homozygotes, which are the only lines to not survive as long as controls, reveal *em2* to most severely accelerate *Lamc2*$^{jeb/jeb}$ symptoms, followed by *em1* then *em15* and *em17*, which are roughly comparable to each other. *Em18* was apparently slightly less deleterious than *em17*, having comparably accelerated disease by ear and tail score measurements but stronger dermal-epidermal adhesion as measured by the tension test, comparable to controls. In cases where *Col17a1* heterozygotes were additionally tested (B6-*Lamc2*$^{jeb/jeb}$ *Col17a1*$^{em/+}$, for *em1*, *em2* and *em8*), they exhibited phenotypes intermediate between homozygotes and B6-*Lamc2*$^{jeb/jeb}$ controls, indicating semi-dominance. Most notably, *em16* did not accelerate *Lamc2*$^{jeb/jeb}$ disease by any method measured, indicating a substantial fraction of extracellular ColXVII can be removed without deleterious effect. Note that the *em16* 2nd cDNA product removes more than *em15*, suggesting *em16* 2$^{nd}$ should be at least as deleterious as *em15*. *Em16* 1$^{st}$ cDNA product deletes a very large segment including much of what is deleted in deleterious *em1*, *em2* and *em15*. This suggests that inclusion of NC5-~distal half of Col3 and NC1 are most critical for maintaining stability and health in *Lamc2*$^{jeb/jeb}$ mice.

Histological evaluation of B6-*Lamc2*$^{jeb/jeb}$ *Col17a1*$^{em1/em1}$ double homozygous mice compared to B6-*Lamc2*$^{jeb/jeb}$ and B6-*Col17a1*$^{em1/em1}$ parental strains at ~29 days old confirmed the pathological diagnosis of JEB and more extensive dermal-epidermal separation than in either

of the single mutants, agreeing with other evidence for disease acceleration when the two genotypes were combined. *B6-Lamc2^{jeb/jeb} Col17a1^{em1/em1}* exhibited dermal-epidermal separation in the trunk, foot, ear and tail skin similar to *B6-Lamc2^{jeb/jeb}* controls but at an earlier age. In addition, the compound mutants had separation at the nail bed leading to nail dystrophy. Additionally tested B6-*Lamc2^{jeb/+} Col17a1^{em1/em1}* at ~29 days old exhibited punctate separations in ear, tail and trunk skin not seen previously in B6-*Lamc2^{jeb/+}* alone [28] or B6-*Col17a1^{em1/em1}* at any age (Fig 6H) indicating that this is a semi-dominant mutation. Notably, while most *B6-Lamc2^{jeb/jeb} Col17a1^{em1/em1}* dermal-epidermal separation not seen in B6-*Lamc2^{jeb/jeb}* age matched controls does appear in them later in life as disease progresses, nail bed separation has not been observed in B6-*Lamc2^{jeb/jeb}* or B6-*Col17a1^{em1/em1}* at any age and is likely a digenic phenotype. The nail phenotype bears similarities to that observed in *em8* (*Lamc2^{wt/wt}*) homozygotes (Fig 7) and humans with *COL17A1* mutation induced JEB nail symptoms [29].

No other unique digenic effects were noted in this strain combination.

## Discussion

Previous work reporting *Col17a1* congenic recombinant modifier fine mapping was apparently made possible by the presence of a *Prdm9* recombination hot spot in the vicinity [26]. The fact that congenic R03L contained two separate recombinant congenics from the parental line initially escaped detection. Once differing genetics were defined across the entire R03L breeding colony, it was made homozygous for the unpublished but more interesting recombinant, between p1277 and p1292, and reanalyzed. It is reported here to further inform the congenic picture. The candidate region is reduced to 723 bp, still containing p1275 and p1277 but notably removing p1292, due to revised R03L mapping information. While results are now available for a strain containing the PWD 'protective' allele of p1292 without p1275 and p1277, the reverse is not. These show that p1292 T/I variance is unable alone to modify *Lamc2^{jeb/jeb}* JEB disease, at least on a B6 background. It does not fully prove that p1275 and p1277 B6/PWD changes provide complete effect without any input from 1292. A separate paper offers evidence of separate p1292 and p1275/1277 modifier effects involving MRL/FVB genetics [14].

The impetus for this project was to create targeted replacements of mouse *Col17a1* loci in B6 mice with PWD alleles to confirm previous strong suspicions that missense changes within the 1084 bp candidate interval were responsible for the disease modifier effects. Replacements were rare events and provided the single lineage *em14* useful for this study. While complicated by the unintended additional p1284 T→S mutation, *em14*+jeb tail tension, ear score and tail score results were all improved compared to B6-*Lamc2^{jeb/jeb}* controls, consistent with the proposed protective roles of Col17a1 p1275 S→G and p1277 N→S changes. While the results do not fully recapitulate the PWD phenotype and several genetic explanations are possible, including involvement of p1292 T→I or other intronic or synonymous changes in the B6/PWD candidate interval, or unidentified off-target effects in the *em14* lineage, the most likely explanation is that the linked p1284 T→S alteration is also affecting L332/COLXVII interaction, and in the opposite direction to the p1275 and p1277 changes. Impact of co-located small deletions *em4*, *em5* and *em9*, as well as localization of the deleterious p1303 R→Q human mutation (mouse p1282) to this interval further verify its importance and the likelihood that the p1284 amino acid change would have an effect.

Despite the relatively modest success of DNA replacement attempts for this project, retention and analysis of many indels generated in founders by TALEN or CRISPR double stranded breaks provides substantial information about distal *Col17a1*, both in terms of resistance to disease induction despite large deletions, and relevance to dermal-epidermal adhesion as

evidenced by modifier effects upon the 'sensitizer' B6-*Lamc2^{jeb/jeb}* mouse model. Lack of dele-terious effects in mice homozygous for 9–18 bp in-frame deletions in *Col17a1* exon 50 on a B6 (*Lamc2^{wt/wt}*) background may not be surprising, despite co-location with evidently important L332/COLXVII interaction loci p1275, p1277, p1284 and p1292, and human disease inducing mutation locus p1303 (mouse p1282), but lack of consistent disease in any of six large indels removing substantial portions of distal COLXVII including apparently the entirety of the hypothesized L332/COLXVII interaction domains in *em1* and *em2* and a substantial portion of the 3'UTR in *em15* is unexpected. Fortunately, the presence of *em8* in the series, with its 10 bp frame-shift deletion expected to lead to protein loss-of-function and accompanying early lethal phenotype when homozygous *Col17a1^{em8/em8}*, similar to that for *Col17a1^{tm1Shzu}* [15], serves as a positive control that knockout of the protein by disruption of this portion of the gene is possible.

As B6-*Lamc2^{jeb/jeb}* modifiers, perhaps the most striking finding of the small indels was that deletion of particular amino acid sets, such as p1275-7 SSN in *em9* and *em10* and p1292-4 TGG in *em4*, *em5* and *em7* produced modifier effects in some indel+jeb combinations but not closely related others. This suggests overall shape changes were more critical than deletions of particular AA.

The tiled large indels give a range of effects from no change (*em16*) to very severe disease acceleration (*em2*) of *Lamc2^{jeb/jeb}* induced disease which can be interpreted as defining the rel-ative importance or contribution of various COLXVII distal extracellular regions to L332/COLXVII connectivity. The *Lamc2^{jeb}* mutation is a hypomorphic allele, resulting in low levels of native L332 protein being made, leading to late onset JEB-nH. This and previous studies indicate that while healthy COLXVII is unable to form even mildly functional dermal-epider-mal binding 'bridges' in the absence of L332, the distal extracellular portion of COLXVII does bind and stabilize L332 [15, 21, 30, 31]. To the extent L332/COLXVII distal binding is dis-rupted, dermal-epidermal connectivity is weakened, but only if L332 is already compromised by its own defects. COLXVII knockouts show that it cannot be totally dispensed with, but apparently the proximal attachments, perhaps between COLXVII and ITGA6B4 which then binds to L332, are most critical. A simplified illustration of how these connections may account for the observed *Lamc2^{jeb/jeb}* modifier phenotypes is included in Fig 11. Diagrams in Fig 11 are based on binary presence or absence of L332/Col17 connections. Variability between *em* lines suggest that multiple connections exist and it is possible to lose or gain a sub-set of them making the phenotype explanations a bit more complex than illustrated here.

*Em17*, *em1* and *em2* encode a series of progressively larger AA deletions with the same end point in mid-NC3, based on the exon 50/51 cDNA boundary. As the deletion size increases, the acceleration effect upon *Lamc2^{jeb/jeb}* disease also increases (*em2* worse than *em1* worse than *em17*). This indicates the COLXVII regions mid-NC6 to mid-NC5, mid-NC5 to mid-Col3 and mid-Col3 to mid-NC3 each have an impact upon L332/COLXVII adhesion and sta-bility. *Em15* acceleration of *Lamc2^{jeb/jeb}* disease is phenotypically similar to *em17*. The similar-ity is likely a coincidence, rather than mapping the effect to the 20 AA region deleted in both, since that region is also deleted in *em16*, which does not accelerate disease. The most likely source of *em15* vs *em16* modifier difference is NC1, which is deleted in *em15* and *em16* cDNA product *em16-2* but not in *em16-1*.

Just as *em1*, *em2* and *em17* share the same cDNA deletion end point at the exon 50/51 boundary, *em16-1*, *em17* and *em18* all share the same cDNA deletion start point exactly half-way through exon 50. Since em17 accelerates *Lamc2^{jeb/jeb}* disease but *em16* does not, it appears that while deletion of mid-Col3 to mid-NC3 has a detrimental effect, *em16*'s additional dele-tion of mid-NC3 through most of Col1 produces a balancing suppressive effect similar to that caused by short deletion *em9* (Fig 3C and 3D).

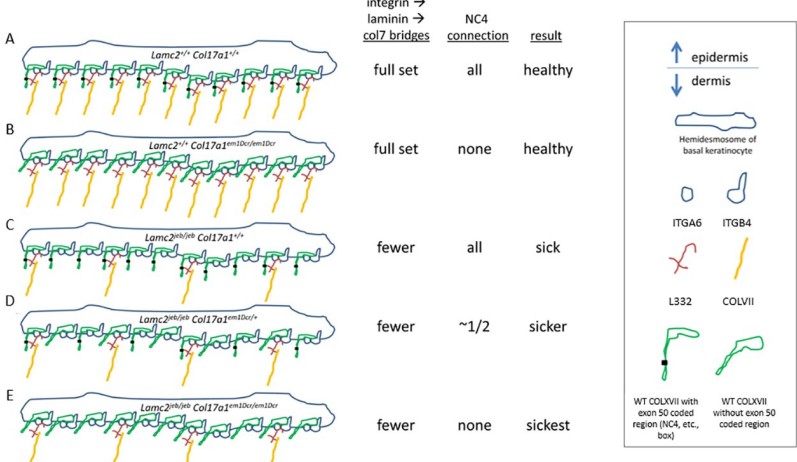

**Fig 11. Proposed mechanical interaction correlation to observed disease phenotypes.** (A) Diagram of 'normal' hemidesmosomal dermal-epidermal connective strands, including components ITGA6, ITGB4, COLXVII, COLVII and L332 including supposed L332/COLXVII supportive connections (black dots). (B) Differs from 'normal' A by absence of L332/COLXVII connections in *Col17a1^{em1/em1}* mice. These mice are healthy. (C) L332/COLXVII connections are present but a shortage of L332 occurs in *Lamc2^{jeb/jeb}* hypomorphic mice, in which the majority of *Lamc2* produced incorporates mutated intron 18 and undergoes nonsense mediated decay before it can incorporate into L332 molecules [28], resulting in fewer L332/COLVII 'bridges' to the dermis. Mice get JEB. (D) *Lamc2^{jeb/jeb}* so fewer L332 are available as in (C) and *Col17a1^{em1/+}* so ~1/2 of the present L332 does not include L332/COLXVII NC4 supportive connections, making the disease worse. (E) *Lamc2^{jeb/jeb}* is worse still when *Col17a1^{em1/em1}* so *none* of the L332 present have L332/COLXVII supportive connections.

Lastly, *em18* deletes the same mid-Col3 to mid-NC3 region as *em17*, but inserts a copy of exon 53, resulting in a Col3/1 hybrid which is about the same size as original Col3, and may restore the function of Col3. This could map the *Lamc2^{jeb/jeb}* disease modifier effect seen by ear and tail deterioration in both *em17* and *em18* to proximal NC3 and not distal Col3. *Em18* connectivity strength as measured by tail tension does not match *em17* –*em17* accelerates disease compared to controls but *em18* does not. This difference apparently maps to the Col1 replacement of part of Col3.

Together *em1*, *em2* and *em15-18* indicate four different regions in distal COLXVII protein which, when deleted in-frame, negatively impact L332/COLXVII binding, not so much as to cause JEB disease alone, but enough to accelerate disease in the presence of a hypomorphic L332, such as that in *Lamc2^{jeb/jeb}* mice: mid-NC6 to mid-NC5, mid-NC5 to mid-Col3, mid-Col3 to mid-NC3 and NC1. *Em16* further implicates deletion of mid-NC3 through most of Col1 as protective compared to wildtype. And *em18* suggests that replacement of part of Col3 with a copy of Col1 can impact dermal/epidermal adhesion strength in a way not directly correlated to disease state.

Together, these results help provide a portrait of how the distal ~1/3 of the extracellular portion of COLXVII interacts in a supportive role with L332. While the proximal portion of COLXVII is necessary for hemidesmosome assembly and proper L332 attachment to ITGA6B4 at the basal keratinocyte face and COLVII in the dermis to provide strong dermal-epidermal connectivity, connection of the distal portion of COLXVII to L332 is not clearly necessary for health when L332 is 'healthy', but the connections apparently do exist, and become consequential when L332 is 'sub-optimal'. This work documents multiple large and small in-frame indels as well as missense SNPs which are innocuous in mice but digenically

impactful in some combinations. This work also highlights the utility of a hypomorphic mouse model such as *Lamc2^{jeb/jeb}* to dissect effects that are otherwise not visible.

These mouse models are relevant to human intermediate JEB in that they begin to define the underlying genetics that explain different clinical subtypes, such as blistering skin with or without nail lesions. They also begin to explain the genetics related to clinical severity of the disease. When comparing patients with similar *LAMC2* mutations but differing symptoms, concurrent polymorphisms in COL17A1 initially interpreted to have no effect may in fact be critical. These observations open up the possibility of focusing on therapeutic approaches not just directed at LAMC2 but also COL17A1 and other genes, as they are discovered.

## Materials and methods

### TALENs and CRISPR/Cas9

Genetic alteration of *Col17a1* sequence in C57BL/6J (B6) mice including replacements and deletions were attempted at four locations involving exon 50. The first three utilized TALEN technology and the last CRISPR/Cas9. Left and right TALENs were designed using ZiFiT [32] to bind and cause double stranded breaks between binding sites `TGTCAGCACTGGGGTCCC` and `TTGAAAAAGAGAAGACCC` (TAL1); `TCACTTAAGGGAGAACTA` and `TGCCCCTTCTGGCTGAGG`, (TAL10, 'site A'); and `TCCTACAGCTCTTCCACG` and `TCACCCAGGGAGCCTCCG` (TAL7, 'site B'). TAL1 targeted intron 50–51 while TAL10 and TAL7 both targeted proximal exon 50, with predicted double-stranded break points at ~1275AA and ~1293AA respectively. Injection cocktails were prepared to include TALEN sets for separate targets (TAL1, TAL7 or TAL10) as mRNA or supercoiled plasmid DNA at 1.5–50 ng/µl each, along with replacement donor templates (S2 Table) provided as ssDNA or super-coiled plasmid DNA at 0–10 ng/µl and RNasin (Promega cat# N2511) [FC = 0.1 U/µl] in MIJ TE Buffer (20µl volume). Subsequently CRISPR/Cas9 with guide targeting `GTTCCGGGATGGCTTCAC` to distal *Col17a1* exon 50 was prepared as cocktails with 200nt ssDNA oligos D57 or G57 (S2 Table) to attempt genetic replacements. All microinjections used fertilized oocytes from C57BL/6J (The Jackson Laboratory JR664) as described previously [33], with CRISPR injections being pronuclear. All injections and transfers of oocytes into pseudo-pregnant females were performed by JAX Microinjection Service. Founders generated were genotyped for various size deletions as well as attempted replacements and those with mutations of interest were bred to B6 and B6-*Lamc2^{jeb/jeb}* (The Jackson Laboratory JR25467) [25] to isolate and characterize. Mutations discussed in this study are named *em1*, *em2*, etc. when homozygous on a B6 background, based on allele names *Col17a1^{em1Dcr}*, etc. When homozygous for both the *Col17a1* mutation and *Lamc2^{jeb/jeb}* (i.e. B6-*Lamc2^{jeb/jeb} Col17a1^{em1/em1}*), they are named *em1+jeb*, etc.

### Replacement sequences

Donor sequences were designed for co-injection with TALEN pairs or CRISPR/Cas9 to serve as templates in repair of double stranded breaks. These templates primarily matched recipient B6 sequence but included select changes meant to be incorporated into an altered genome. For TAL1, the donor was 4034 nucleotides long and incorporated PWD alleles at 50 loci in a ~2900 bp segment extending from *Col17a1* exon 49 through the 3'UTR flanked by ~600 bp ends with complete homology to the native B6 sequence. For TAL7, a 200nt donor was designed to convert amino acid 1282 R→Q to imitate the R1303Q JEB disease causing mutation in humans [17]. For TAL10, a 53nt donor incorporating two SNPs to cause amino acid changes 1275 S→G and 1277 R→S was designed to replace B6 with the PWD allele at those

two loci only. Each donor also included 2–3 SNP changes in one of the TALEN binding sites intended to prevent TALEN cleavage after successful integration. These changes were selected so as to not alter the resulting amino acid sequence. Additional donors were designed to 'humanize' a region in distal exon 50 when targeted by CRISPR/Cas9. Donors were primarily injected as single stranded DNA though some TAL1 trials used supercoiled plasmid DNA. Complementary ssDNA donor templates were made for TAL7 and TAL10 and injected in separate experimental sets. These donor sequences are listed in S2 Table. Full TALEN sequences are in S3 Table.

## Genotyping

Primers for PCR were designed using Primer Express 2.0 software (Applied Biosystems, Inc.) to a 58–60°C annealing temperature. Thermocycling was typically 40 cycles of 94°C for 30 seconds, 60°C for 1 minute and 70°C for 1 minute. Products were run on a 0.7% agarose plus 1.5% Synergel gel containing Ethidium Bromide for size discrimination. DNA for genotyping and sequencing was obtained from retro-orbital blood washed in Buffone's buffer and digested in Proteinase K. Primers were designed across the TALEN and CRISPR targeted regions to give products of 121 bp to 1825 bp and to detect deletions of ~4 bp to ~1500 bp in size, including those skewed to one side or the other of each cut site.

*Em1-2*: Initial detection of the 567 bp deletion mutant *Col17a1^(em1Dcr)* (*em1*) and subsequent tracking was performed using TAL1ComF (GCAAAGTACCAAGCCCTTTGC) and 723v6R (AAGGGCCTCAAAGGTCACTCTA, 711 bp wt, 144 bp mutant). Zygosity testing included a second reaction of TAL1ComF with 723v3R (GAGCCTATGGTGCAGGAGATG, 419 bp wt, no band for mutant). Identification and tracking of the 650 bp deletion mutant *Col17a1^(em2Dcr)* (*em2*) was performed using 723F (TTCAATCCCCACCCATCTGAT) and 724R (AGATATCGCCTGAAAGGGACAA, 1084 bp wild-type, 434 bp mutant). Zygosity was tested using a second reaction of TAL7ComF (TCTCCTGCACCATAGGCTCC) with 723v4R (TCCTGATGTTCGCAGCTTCAT, 201 bp wildtype, no band for mutant).

*Em4-13*: TAL7ComF (TCTCCTGCACCATAGGCTCC) and TAL7ComF2 (CACCATTGGTGCCATATATACCAC) were each combined with 723v4R (TCCTGATGTTCGCAGCTTCAT) to generate 201 and 284 bp wild-type products respectively. TAL7ComF was also combined with 723v7R (TGGAGATGGTCACTTAAGGGAGAA) to give a 141 bp wild-type product. 9–18 bp deletions at both TAL7 (*em4*, *em5*, *em7*) and TAL10 (*em9*, *em10*, *em13*) cut sites were identified and tracked using these markers. 284 bp PCR products were also sequenced to identify smaller deletions.

*Em14*: 1275AA replacement *em14* genotyping was performed by pairing 1275Repl (AAGTGCCCCTTCTGGCGCT) or 1275B6 (AAGTGCCCCTTCTGGCTGA) primers with 723v4R to give replacement or B6 specific 125 bp bands.

*Em15-16* are both detected using Afg57F1 (GCAGGTGGTGGTATATATGGCAC) and Afg57R5 (CAAGTCTCATATTAAACATTGCAGCTATTC) which produces an 1825 bp wild-type amplicon. Zygosity is determined by PCR using Afg57F1 coupled with Afg57R2com (GCTCCTTCCCCCTCTGAGG), which gives a 144 bp wild-type product, as Afg57R2com resides within the deleted region of each.

*Em17-18* are both detected using Afg57F1 (GCAGGTGGTGGTATATATGGCAC) with Afg57R3 (GAAGGGCAGTGTCCTCGGA), which give a 558 bp wild-type product.

*Lamc2* was genotyped using *D1Dcr2*, *D1Dcr15*, *Lamc2Dcr1a* and *Lamc2Dcr1b* as previously published as well as *D1Dcr44* and *D1Dcr55*, newly identified as within the B6.129X1-*Lamc2^(jeb/jeb)* congenic interval. Chr19 SSLP primers to distinguish B6 vs PWD were previously published [25, 34].

## Mice

All TALEN potential founder mice were genotyped by PCR. TAL7 and TAL10 mice were also genotyped by sequencing from 284 bp products bracketing the TALEN cut sites or using other primers, as appropriate. Select identified founders were bred to B6 and then intercrossed to make each line homozygous isogenic. One line, *Col17a1^em8Dcr^*, was found to be neonatal lethal in the homozygous state and was primarily maintained in het x het matings. All lines were also bred to B6.129X1-*Lamc2^jeb/jeb^* (B6-*Lamc2^jeb/jeb^*, JR25467) previously described [25, 34] to produce mice homozygous for the *Lamc2^jeb^* mutation and heterozygous or homozygous for each characterized *Col17a1* mutation. C57BL/6J used as breeders and experimental controls were obtained from The Jackson Laboratory, Bar Harbor, Maine. All information on mice, including breeding and experimental data was collected in a customized copy of JAX Colony Management System (JCMS) [35]. All animal procedures were approved by IACUC (approval number 01022). C57BL/6J-*Col17a1^em8Dcr^* is publicly available from The Jackson Laboratory as strain 33908. Most other C57BL/6J-*Col17a1^emXDcr^* lines were privately sperm frozen and are available for cryorecovery from The Jackson Laboratory (see S4 Table for strain IDs). All mice used for breeding or experiments that required euthanasia were euthanized by $CO_2$ asphyxiation. None of our procedures required anesthesia or analgesia.

## B6.PWD correction of R03L

B6.PWD congenic mouse strains R03F and R03L, each homozygous *Lamc2^jeb/jeb^* and for PWD chr19 congenic segments with recombinant break points within *Col17a1*, were previously described [25]. The R03L intra-*Col17a1* break point was subsequently determined by sequencing of additional animals to be in error and is corrected here (Fig 1). B6.PWD strains R03F and R03L were privately sperm frozen and are available for cryorecovery from The Jackson Laboratory (see S4 Table for strain IDs).

## Ear and tail scoring and tail tension testing

Male experimental mice were aged and observed weekly and ear and tail scores ranging from 0 (unaffected) to 6 ('severely affected') were recorded as previously described [25]. Male mice were euthanized by $CO_2$ asphyxiation followed by thoracic puncture and then tail tension tested at indicated ages as previously described [34].

## Statistical analysis

As with previous, ear and tail score significance is determined using survival comparisons of age at which mice first reach scores of '4' (moderately affected) with censored values when they do not reach a score of '4' before being removed from the experiment. Due to the large number of comparisons necessary, a search algorithm was created in JCMS such that, upon selection of a strain of interest, it would identify male mice in 'jebaging' experiments and separately determine age in weeks when ear and tail scores first reach 4 and add to lists with censor '0' or age at death with censor '1' if never reached score '4'. Columns with age in weeks and censor were pasted from here into GraphPad Prism in which they were pairwise survival compared using Log-rank (Mantel-Cox) and Gehan-Breslow-Wilcoxon tests, with the less significant of the two being indicated on graphs and reported.

All tension statistics are 1-way ANOVA using Tukey's multiple comparisons test in GraphPad Prism.

## RNA and cDNA

Strips of skin totaling ~2" x 1/8" were collected from euthanized mice in 4 ml RNAlater, stored at room temperature for 24 hours and then frozen at -40˚C until ready to process. Total RNA was isolated using the standard Trizol reagent method (Invitrogen) and processed to make cDNA using the MessageSensor™ RT kit (Ambion/Applied Biosystems) by JAX Gene Expression Service.

## Sequencing

PCR products from DNA and cDNA were magnetic bead purified and sequenced on an Applied Biosystems 3730xl by the JAX DNA Sequencing Service. Sequence data were analyzed using Applied Biosystems Sequencing Analysis software version 5.2 and Sequencher software version 4.10.

## Histology

Female B6-*Lamc2*$^{jeb/jeb}$ homozygotes, B6-*Col17a1*$^{em1Dcr/em1Dcr}$ homozygotes, B6-*Lamc2*$^{jeb/+}$ *Col17a1*$^{em1Dcr/em1Dcr}$ (het for *Lamc2*, home for *Col17a1*), B6-*Lamc2*$^{jeb/jeb}$ *Col17a1*$^{em1Dcr/em1Dcr}$ double homozygotes and B6 wild-type controls were euthanized by $CO_2$ asphyxiation at 26–30 days of age (n = 2–4 of each) and necropsied. Female B6-*Col17a1*$^{em1Dcr/em1Dcr}$ and B6-*Col17a1*$^{em2Dcr/em2Dcr}$ homozygotes and B6 wild-type controls were additionally necropsied at 10 months +/- 2 weeks of age (n = 4–13 of each) and subjected to a more thorough tissue collection and examination. Skin samples were also collected and evaluated from n = 2 B6-*Col17a1*$^{em15Dcr/em15Dcr}$ males and n = 4 B6-*Col17a1*$^{em16Dcr/em16Dcr}$ males at 52 weeks of age. Samples were also collected from n = 1–2 B6-*Col17a1*$^{em8Dcr/em8Dcr}$ and B6-*Col17a1*$^{em8Dcr+}$ each at 6, 10 and 27 days old. Tissues collected and examined at 4 and 52 weeks of age were skin from the ears, tail, foot (including foot pads and nail unit), and dorsal skin. Tissues collected from *em8* at 6–27 days old were the same plus the entire head to evaluate the tongue and nasopharynx *in situ*. Tissues collected and examined at 10 months of age were from the following: Swiss rolls of the duodenum, jejunum, ileum, and colon (with anus and perineal skin), longitudinal section of the stomach with esophagus and cecum (inflated with fixative), cross sections of the left lateral and medial lobes of liver to include the gall bladder, spleen, left and right kidneys with adrenal glands, reproductive organs (ovary, uterine tube, uterus, mammary glands), clitoral gland, salivary gland cluster with cervical lymph nodes, heart, esophagus and trachea with thyroid and parathyroid glands, tongue, longitudinal sections out of the center of the lobes of both lungs, dorsal skin, ear skin (pinna), ventral skin, muzzle skin, eyelid, longitudinal section of hind leg including stifle/knee joint, longitudinal section of front leg including shoulder and elbow joints, longitudinal section of hind foot (soft tissues, bone, and nail unit/footpad), longitudinal section of front foot (soft tissues, bone, and nail unit/footpad), longitudinal section and cross section of the lumbar spine, longitudinal section and cross section of the tail and sections of the lower jaw. These were collected and fixed in Fekete's acid alcohol formalin overnight, after which they were transferred and stored in 70% ethanol. Bones were processed in Cal-Ex® (Fisher, Pittsburgh, PA). The cervical spine and skull with brain were collected in Bouin's solution. The skull was cut longitudinally and perpendicularly to provide sections of brain and all bone and soft tissues in the region including the eye. Pancreata were collected in Bouin's solution and stained with aldehyde fuchsin. Pancreata were also collected and fixed in Fekete's solution with the intestinal rolls. Tissues were then trimmed and embedded in paraffin, cut into 6 μm sections, stained with hematoxylin and eosin (H&E) [36] and examined by an experienced pathologist (JPS).

## Capillary immunoelectrophoresis

Trunk skin from 3–4 day old pups (typically females) homozygous for various B6-*Col17a1*$^{em/em}$ or B6 as a control was flash frozen in liquid nitrogen and stored at -80˚C until processed to extract protein. Previously frozen juvenile mouse skin tissues were weighed and then pulverized in TT1 tissueTUBEs using the CP02 cryoPREP Impactor, (Covaris. Inc., Washburn, Mass.). Five µl of ice cold RIPA buffer (150 mM NaCl, 1.0% IGEPAL® CA-630, 0.5% sodium deoxycholate, 0.1% SDS, 50 mM Tris, pH 8.0) was added per mg of tissue to Miltenyi M Tubes on ice. The pulverized tissues were quickly added and shaken down before being homogenized in a gentleMACS Dissociator, (Miltenyi Biotec Inc., Auburn, CA). The foam was reduced by centrifugation for 5 minutes at 2k x g before the lysates were transferred to 2 ml microtubes and centrifuged for 10 minutes at 21k x g. The supernatants were removed and each were frozen @ -80˚C in several aliquots. One aliquot of each mouse strain was used for protein concentration determination using the Micro BCA assay kit, (Thermo Scientific, Rockford, Ill.)

Subsequent aliquots of the skin lysates were defrosted on ice and diluted to 0.4 mg/ml in 1% SDS. Collagen XVII quantitation was determined for each lysate using HPA043673 rabbit polyclonal (Atlas Antibodies, Stockholm, Sweden) and ERP18614 rabbit monoclonal (abcam ab184996, Cambridge, MA) antibodies, which bind proximal and distal Collagen XVII respectively. Each was diluted 1:50 and run in separate capillaries on the WES immunoassay system (ProteinSimple, San Jose, CA). Anti-Vinculin mouse monoclonal antibody MAB6896 (R&D Systems, Minneapolis, MN) was also included in each primary antibody mix as a loading control. Protein Simple's "ready-to-use" (RTU) HRP conjugated secondary antibodies, anti-rabbit and anti-mouse, were used as a 50:50 mix. The standard run protocol was used except the primary incubation time was extended to 120 minutes followed by three washes.

## Supporting information

**S1 Table. Breeding performance of *Col17a1*$^{em/em}$ lines when *Lamc2*$^{wt/wt}$.**
(XLSX)

**S2 Table. Replacement sequences.**
(DOCX)

**S3 Table. Full TALEN sequences (Joung #s DR-TAL-0727-28) for intron 50 cut (TALEN1).**
(DOCX)

**S4 Table. Private strains used in this study.**
(XLSX)

**S1 Fig. Raw gel image from which Fig 7B gel image was derived.**
(BMP)

**S1 File. Additional supporting information.**
(XLSX)

## Acknowledgments

We thank V. Kennedy for assistance with initial sample preparation.

## Author Contributions

**Conceptualization:** Thomas J. Sproule, Derry C. Roopenian, John P. Sundberg.

**Data curation:** Thomas J. Sproule, Robert Y. Wilpan, Benjamin E. Low, Kathleen A. Silva, Deepak Reyon.

**Formal analysis:** Thomas J. Sproule, Robert Y. Wilpan, Derry C. Roopenian.

**Funding acquisition:** Derry C. Roopenian, John P. Sundberg.

**Investigation:** Thomas J. Sproule, Kathleen A. Silva, Deepak Reyon.

**Methodology:** Thomas J. Sproule, Benjamin E. Low, Deepak Reyon, J. Keith Joung, Michael V. Wiles.

**Project administration:** Derry C. Roopenian, John P. Sundberg.

**Resources:** Deepak Reyon, J. Keith Joung, Michael V. Wiles, Derry C. Roopenian.

**Supervision:** Derry C. Roopenian.

**Writing – original draft:** Thomas J. Sproule, Derry C. Roopenian, John P. Sundberg.

**Writing – review & editing:** Thomas J. Sproule, Robert Y. Wilpan, Benjamin E. Low, Kathleen A. Silva, Deepak Reyon, J. Keith Joung, Michael V. Wiles, Derry C. Roopenian, John P. Sundberg.

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
