## [Decision Letter · Decision Letter 0]

24 Aug 2023

PONE-D-23-17953Functional analysis of Collagen 17a1: a genetic modifier of Junctional Epidermolysis Bullosa in micePLOS ONE

Dear Dr. Sproule,

Thank you for submitting your manuscript to PLOS ONE. After careful consideration, we feel that it has merit but does not fully meet PLOS ONE’s publication criteria as it currently stands. Therefore, we invite you to submit a revised version of the manuscript that addresses the points raised during the review process.

We look forward to receiving your revised manuscript.

Kind regards,

Gerhard Wiche, Ph.D.

Academic Editor

PLOS ONE

2. To comply with PLOS ONE submissions requirements, in your Methods section, please provide additional information regarding the experiments involving animals and ensure you have included details on methods of anesthesia and/or analgesia

“TJS, RYW, BEL, KAS, DR, JKJ, MVW, JPS and DCR have no competing interest. JKJ has, or had during the course of this research, financial interests in several companies developing gene editing technology: Beam Therapeutics, Blink Therapeutics, Chroma Medicine, Editas Medicine, EpiLogic Therapeutics, Excelsior Genomics, Hera Biolabs, Horizon Discovery, Monitor Biotechnologies, Nvelop Therapeutics (f/k/a/ ETx, Inc.), Pairwise Plants, Poseida Therapeutics, SeQure Dx, Inc., Transposagen Biopharmaceuticals, and Verve Therapeutics. JKJ’s interests were reviewed and are managed by Massachusetts General Hospital and Mass General Brigham in accordance with their conflict of interest policies. JKJ is a co-inventor on various patents and patent applications that describe gene editing and epigenetic editing technologies.”

Please include your updated Competing Interests statement in your cover letter; we will change the online submission form on your behalf

6. We notice that your supplementary Tables are included in the manuscript file. Please remove them and upload them with the file type 'Supporting Information'. Please ensure that each Supporting Information file has a legend listed in the manuscript after the references list.

Reviewers' comments:

Reviewer's Responses to Questions

**Comments to the Author**

1. Is the manuscript technically sound, and do the data support the conclusions?

Reviewer #1: Yes

2. Has the statistical analysis been performed appropriately and rigorously? 

Reviewer #1: Yes

3. Have the authors made all data underlying the findings in their manuscript fully available?

Reviewer #1: Yes

4. Is the manuscript presented in an intelligible fashion and written in standard English?

Reviewer #1: Yes

5. Review Comments to the Author

Reviewer #1: This paper builds upon the authors' groundbreaking work on the modifier effects of Col17a1 for lamc2 hypomorphic mice (Sproule et al., PLoS Genet 2014). The authors delve deeper into the production of multiple Col17a1 mutants, further confirming the significance of the COL17 C-terminal region. Additionally, this study illuminates and explains the pathomechanisms of human junctional EB with the COL17A1 p.R1303Q variant. Despite the content's lack of a biochemical approach (e.g., protein-protein binding assay), the research paves the way for further elucidation of human and mouse basement membrane biology. I have only a few comments, which are as follows:

1) The nomenclature of epidermolysis bullosa in the paper is outdated (e.g., JEB, Herlitz type -> JEB, severe). It would be preferable to adhere to the current classification (Has et al., Br J Dermatol 183:614-327, 2020) rather than the former one (ref. 8).

2) The depiction of the em8 deletion in Fig. 2A is too minuscule to discern.

3) In Fig. 10, the sequence is listed as "integrin -> laminin -> col7 bridges." Should this be "col17" instead of "col7"?

6. PLOS authors have the option to publish the peer review history of their article (what does this mean?). If published, this will include your full peer review and any attached files.

Reviewer #1: No

---

## [Author Response · Author response to Decision Letter 0]

10 Sep 2023

Thank you for your thoughtful comments concerning our submitted manuscript PONE-D-23-17953

Functional analysis of Collagen 17a1: a genetic modifier of Junctional Epidermolysis Bullosa in mice. 

Journal requirements.

“1. Please ensure that your manuscript meets PLOS ONE's style requirements, including those for file naming.”

Response: This has been done.

“2. To comply with PLOS ONE submissions requirements, in your Methods section, please provide additional information regarding the experiments involving animals and ensure you have included details on methods of anesthesia and/or analgesia.” 

Response: Two sentences were added to the Mice subsection of Materials and Methods to clarify that all mice used by us that did not die by natural causes were euthanized by CO2 asphyxiation and that none of our procedures required anesthesia or analgesia.

“3. We note that the grant information you provided in the ‘Funding Information’ and ‘Financial Disclosure’ sections do not match.”

Response: I have reviewed and confirmed the grant numbers in the Funding Information section. The discrepancies you refer to may be due to the limitations of the Funding Information pull-down lists. In Financial Disclosures we mention two recipients for each DeBRA grant but Funding Information only allows us to list one. Also DeBRA UK is not on your institute pull-down list, so I selected DeBRA International. If there are other discrepancies I have overlooked, please let me know and I will try to help resolve them.

4. Competing Interests statement.

Response: The modified Competing Interests statement is included in the cover letter. Please change the online submission to match it.

5. In your cover letter, please note whether your blot/gel image data are in Supporting Information or posted at a public data repository, provide the repository URL if relevant, and provide specific details as to which raw blot/gel images, if any, are not available.

Response: This manuscript includes one gel image. The original of that has been added to the online submission as Supporting Information and that has been stated in the cover letter.

“6. We notice that your supplementary Tables are included in the manuscript file. Please remove them and upload them with the file type 'Supporting Information'. Please ensure that each Supporting Information file has a legend listed in the manuscript after the references list.”

Response: All supplemental tables have been removed from the Manuscript and submitted as Supporting Information files. Legends for supplemental Tables in the Manuscript have been moved to after References.

7. Please review your reference list to ensure that it is complete and correct.

Response: This has been done.

Reviewer 1 comments.

1.The nomenclature of epidermolysis bullosa in the paper is outdated (e.g., JEB, Herlitz type -> JEB, severe). It would be preferable to adhere to the current classification (Has et al., Br J Dermatol 183:614-627, 2020) rather than the former one (ref. 8).

Response: Text has been reworded to replace mentions of Herlitz JEB with ‘severe JEB’ and non-Herlitz JEB with ‘intermediate JEB’.

2.The depiction of the em8 deletion in Fig. 2A is too minuscule to discern.

Response: Fig 2 has been split to two figures and enlarged to make it more readable. Following figure numbers have been incremented to compensate.

3.In Fig. 10, the sequence is listed as "integrin -> laminin -> col7 bridges." Should this be "col17" instead of "col7"?

Response: The labeling is correct. Previous work indicates that integrin alpha 6, integrin beta 4 and collagen 17 are all bound to each other, with the molecules partially inside hemidesmosomes of the epidermis and partially outside. The integrins then both bind to laminin 332. Collagen 17 also apparently binds to laminin 332 in normal cases, but not in some of our mutants. Laminin 332 then binds to collagen 7 in the dermis, completing the ‘bridges’ which holds skin together. Figure 11 legend has been changed to clarify this.

---

## [Editor Report · Decision Letter 1]

21 Sep 2023

Functional analysis of Collagen 17a1: a genetic modifier of Junctional Epidermolysis Bullosa in mice

PONE-D-23-17953R1

Dear Dr. Sproule,

We’re pleased to inform you that your manuscript has been judged scientifically suitable for publication and will be formally accepted for publication once it meets all outstanding technical requirements.

Kind regards,

Gerhard Wiche, Ph.D.

Academic Editor

PLOS ONE
---

## [Editor Report · Acceptance letter]

25 Sep 2023

PONE-D-23-17953R1 

Functional analysis of Collagen 17a1: a genetic modifier of Junctional Epidermolysis Bullosa in mice 

Dear Dr. Sproule:

I'm pleased to inform you that your manuscript has been deemed suitable for publication in PLOS ONE. Congratulations! Your manuscript is now with our production department. 

Kind regards, 

on behalf of

Prof. Gerhard Wiche 

Academic Editor

PLOS ONE